# MRS-YOLO : A YOLO MODEL FOR SIGNAL DETECTION IN MULTI-RESOLUTION SPECTROGRAMS

## ABSTRACT

Many real-world signals contain structures spanning multiple time–frequency (TF) scales, where short transients and long-duration patterns coexist. Standard spectrograms, based on the short-time Fourier transform, are constrained by the Heisenberg uncertainty principle, which here translates into the well-known trade-off between time and frequency resolutions. We propose MRS-YOLO, a multi-resolution extension of YOLO that processes spectrograms at complementary scales through parallel branches and fuses them with an attention block. On a challenging datasets of heterogeneous radio-frequency signals with spectral congestion, low SNR, and stealthy emissions, MRS-YOLO achieves higher recall in low-SNR regimes and stronger classification accuracy than single-resolution baselines, demonstrating the value of explicit multi-scale representation learning in TF analysis.[1]

## 1 INTRODUCTION

Many real-world signals exhibit structures that unfold across multiple resolutions in time and frequency. Short transients and long-duration broadband patterns often coexist, making it difficult for fixed-resolution models, originally designed for natural images, to capture all relevant information. This challenge is particularly evident in *time–frequency (TF) analysis*, which underpins domains as diverse as speech and audio processing, medical diagnostics, radar and sonar, and spectrum monitoring. In each case, one must represent signals that simultaneously demand fine temporal localization and precise spectral characterization.

The short-time Fourier transform (STFT) is one of the most widely used TF representations, valued for its intuitive interpretation and efficient computation. However, it suffers from the well-known time–frequency resolution trade-off, which is a manifestation of the Heisenberg uncertainty principle: short windows give fine temporal localization but blur frequency information, while long windows sharpen frequency resolution at the cost of temporal precision. Other transforms, such as wavelet transforms, Wigner–Ville distributions, or more general Cohen's class representations, offer alternative ways of balancing resolution, but all remain fundamentally constrained by this uncertainty principle (Mallat, 1999). Consequently, no single TF representation allow to capture complex structure of signals.

From a representation learning perspective, the challenge of *multi-resolution* detection in time–frequency analysis strongly resembles the well-studied problem of *multi-scale* learning in vision (Lindeberg, 2013). Yet the two are not equivalent. In vision, multi-scale methods address object size variability within a single image, typically by reusing features across scales. In contrast, multi-resolution TF analysis relies on multiple spectrograms computed with different STFT windows. As a result, a signal may be invisible at one resolution but clearly detectable at another, making cross-resolution reasoning essential.

Signal detection further requires identifying not only the presence of an event but also its time–frequency extent and class label. This is precisely the goal of *object detection* frameworks, and among them, YOLO is particularly attractive thanks to its single-stage design and its ability to handle multi-scale variation through feature pyramids. Recent studies have extended YOLO beyond vision and applied it to spectrograms in domains such as RF signal analysis (Zhu et al., 2024;

---

[1]Code available at `https://github.com/ICLRanonymous2026/MRS_YOLO_ICLR26`.

Sarkar et al., 2024; Ma et al., 2024), bioacoustics (Parcerisas et al., 2024), and seismology (Xu et al., 2023). By treating spectrograms as images, these works successfully detect localized signal events. Yet, they all rely on a single fixed-resolution representation, thereby assuming that one can select in advance the "right" time–frequency scale for the signals of interest, a strong assumption that rarely holds in practice.

In this work, we introduce MRS-YOLO, a multi-resolution extension of YOLO for time–frequency object detection. Our contributions are fourfold: 1. We redesign the YOLO backbone to process multiple spectrograms in parallel, enabling joint reasoning across complementary resolutions. 2. We introduce a dedicated fusion module that integrates multi-resolution features into a coherent representation. 3. We incorporate a lightweight time–frequency attention block to enhance salient patterns while keeping inference efficient. 4. We establish a challenging RF detection benchmark with heterogeneous signals and SNRs, against which MRS-YOLO achieves substantial gains over single-resolution baselines.

## 2  RELATED WORK

**Multi-resolution Signal Representations.**    Recent progress in deep learning has aimed to mitigate the limitations of conventional single-resolution spectrogram analysis for TF signal detection. Two complementary research paths have emerged in response.

One approach discards spectrograms altogether and processes raw in-phase and quadrature (I/Q) samples directly. STFNet (Yao et al., 2019), for example, employs a collection of trainable Fourier kernels that adaptively capture frequency-selective patterns. Other studies design multi-scale convolutional backbones for I/Q streams, using dilations or progressively larger kernels to extract features across multiple resolutions (Cui et al., 2024; Chen et al., 2019). Hybrid architectures such as IQ-Former (Shao et al., 2025) go a step further by combining a time-domain branch with learnable spectro-temporal modules, allowing the network to fuse fine-grained local cues with broader contextual information.

A second strategy explicitly generates several spectrogram views using different STFT window lengths. These complementary representations are then processed jointly by neural networks. For instance, SLNet (Li & Zhou, 2023) constructs spectrograms with 64-, 256-, and 1024-sample windows and applies an attention gate to modulate their contributions, leading to strong improvements in Wi-Fi gesture recognition. In another example, (Lee & Oh, 2020) leverage multiple spectrogram resolutions within a hybrid CNN–RNN framework to enhance the detection of frequency-hopping signals while mitigating leakage and resolution artifacts.

Together, these works highlight a growing trend toward multi-resolution signal representations, either learned directly from I/Q waveforms or derived through multi-window spectrograms, as a way to better capture the diverse temporal and spectral structures that characterize complex RF environments. Yet, these approaches have primarily been applied to classification or specialized sensing tasks, leaving open the question of how to integrate multi-resolution reasoning into efficient detection frameworks.

**YOLO-based Detectors.**    Object detection models are commonly divided into two categories: two-stage detectors (Cai & Vasconcelos, 2018; He et al., 2017; Ren et al., 2015), which generate region proposals before classification, and one-stage detectors, which perform dense predictions in a single pass. Among the latter, the YOLO series (Redmon et al., 2016; Terven et al., 2023) has become a leading framework due to its speed and accuracy. Recent YOLO versions have introduced incremental innovations: YOLOv8 (Jocher et al., 2023) adopts anchor-free heads, C2f blocks, and Distribution Focal Loss; YOLOv9–11 (Wang & Liao, 2024; Ao Wang, 2024; Jocher & Qiu, 2024) improve architecture, training, and inference with modules like GELAN, PGI, C3K2, and C2PSA. Meanwhile, attention-based detectors such as DETR (Carion et al., 2020) have demonstrated strong global context modeling but remain impractical for real-time tasks. To address this, YOLOv12 (Tian et al., 2025) introduces Area Attention, achieving enhanced accuracy with minimal computational overhead.

Beyond the core YOLO architecture, attention mechanisms have increasingly been integrated to boost detection performance.  ViT-YOLO (Zhang et al., 2021) incorporates Multi-Head Self-

Attention (MHSA) in its Darknet-based backbone and a BiFPN neck, enabling it to capture long-range dependencies and perform effective cross-scale feature fusion. Other efforts combine spatial and channel attention to jointly model spatial structure and inter-channel dependencies. For instance, CBAM (Woo et al., 2018) introduces a lightweight convolutional block attention module that sequentially applies channel and spatial attention, and it has also been integrated into YOLO-based architectures (Hu et al., 2021; Yan et al., 2024). In the context of RF signal target detection, Ma et al. (2024) enhanced YOLOv8 by inserting a CBAM right before the SPPF layer. More recently, SCCA-YOLO (Wei & Wang, 2025) replaced the standard C2f blocks of YOLOv8 with a Spatial-Channel Collaborative Attention (SCCA) module, composed of a Shared Multi-Semantic Spatial Attention (SMSA) and a Progressive Channel-wise Self-Attention (PCSA) sub-module (Si et al., 2025).

Collectively, these developments illustrate the effectiveness of enriching YOLO with attention-based context modeling while retaining real-time efficiency. However, despite the growing use of attention in visual detection, existing YOLO variants have not yet explored attention mechanisms specifically designed for spectrogram data and time–frequency patterns, which are central to RF signal detection.

**Time–Frequency Attention.** Motivated by the above gap, we review time–frequency (TF) attention modules. Zhang et al. (2022) introduce a compact module that learns a 2D attention mask over spectrograms, enabling a ResTCN system to emphasize salient TF regions for speech enhancement with negligible added complexity. In a different setting, Lin et al. (2022) show that selectively weighting informative channels, frequency bands, and time segments within a CNN improves automatic modulation recognition. Ding et al. (2022) propose a TF Transformer with a dedicated tokenizer and encoder to extract discriminative patterns from vibration spectrograms, boosting fault diagnosis accuracy, while Mu et al. (2021) decouple temporal and spectral attention in a TFCNN to better capture structure in environmental sound classification.

Although effective, many TF-attention designs increase computational load. To address efficiency, Cai et al. (2024) present Time–Frequency Separate Convolutions (TF-SepConvs), which decouple temporal and spectral processing and employ depthwise separable convolutions to reduce parameters and inference cost. To the best of our knowledge, these TF-attention mechanisms have not yet been instantiated within YOLO-style detectors, highlighting an opportunity to couple multi-resolution spectral inputs with lightweight TF attention in real-time detection frameworks.

## 3 METHODOLOGY

In this section, we present **MRS-YOLO**, an architecture tailored for robust TF signal detection from spectrograms. The model processes multiple spectrograms computed at different resolutions using dedicated convolutional branches, whose features are subsequently fused and integrated into a standard YOLO neck and head. We describe the generation of multi-resolution spectrograms, the design of per-branch backbones, the cross-resolution fusion strategy, and our Time–Frequency Attention (TF-Attn) module that we apply pervasively.

### 3.1 MULTI-RESOLUTION SPECTROGRAMS

The short-time Fourier transform (STFT) provides a localized representation of a discrete-time signal $x[n]$, $n = 0, \ldots, N-1$. At analysis scale $r$, with window function $w_r[n]$, length $L_r$, and hop size $h_r$, we form the spectrogram

$$S^{(r)}[m,k] = \left| \sum_{n=0}^{L_r-1} x[n + m\, h_r]\, w_r[n]\, e^{-j\, 2\pi kn/L_r} \right|^2, \qquad \mathbf{S}^{(r)} \in \mathbb{R}^{H_r \times W_r}, \qquad (1)$$

where $m$ and $k$ index the time frames and frequency bins.

Because short windows emphasize temporal detail but blur spectral content, while long windows achieve the opposite, no single pair $(L_r, h_r)$ suffices. We therefore construct a bank of normalized spectrograms at multiple resolutions, $\mathcal{S} = \{\, \mathbf{S}^{(r)} \,\}_{r=1}^{R}$, each interpreted as a single-channel (grayscale) image and processed by a dedicated convolutional branch within MRS-YOLO, enabling the network to exploit complementary information across scales.

## 3.2 Multi-Resolution Backbone and Fusion

We denote by $P_4$, $P_5$, and $P_6$ the backbone feature maps at progressively coarser spatial scales ($64\times64$, $32\times32$, and $16\times16$, respectively). Each branch backbone maps its input spectrogram $\mathbf{S}^{(r)}$ to an aligned feature map $P_4^{(r)} \in \mathbb{R}^{C\times64\times64}$, ensuring that discriminative spectro–temporal structure is preserved. Alignment is performed using a *stride schedule*, a sequence of anisotropic resolution-changing steps $(s_F, s_T)$ along frequency and time. If one axis is below the target size of 64, it is upsampled while the other axis is left unchanged. If the map is anisotropic, only the longer axis is downsampled until both dimensions become comparable. Once isotropy is reached, symmetric strides $(2, 2)$ are applied until the target $64\times64$ resolution is obtained.

At the $P_4$ level, we fuse the per-branch feature maps by concatenation along the channel dimension. While concatenation preserves information from all resolutions, it also increases the channel width proportionally to the number of branches, leading to higher computational cost in subsequent layers. To control this, we apply a pointwise convolution that compresses the channel dimension back to $C$. Before this compression, we insert a **Spatial–Channel Synergistic Attention (SCSA)** block Si et al. (2025), which refines the concatenated tensor by enhancing spatial structures through SMSA (Shared Multi-Semantic Spatial Attention) and channel dependencies through PCSA (Progressive Channel-wise Self-Attention). This ensures that the most informative cross-resolution cues are emphasized before dimensionality reduction.

From this fused representation, the backbone continues with the downsampling schedule to produce $P_5$ and $P_6$. The $P_6$ feature map is further refined with **SPPF** (He et al., 2015) and **C2PSA** modules from YOLOv11 (Jocher & Qiu, 2024). The resulting multi-scale set $(P_4, P_5, P_6)$ forms the input to the YOLO neck, which aggregates features across scales, and is finally passed to the standard YOLOv11 detection head for joint localization and classification. The overall achitecture is presented in Figure 1.

## 3.3 Time–Frequency Attention (TF-Attn)

Inspired by Cai et al. (2024), we introduce TF-Attn as a lightweight mechanism designed to enhance spectro–temporal patterns while keeping computation affordable. The block relies on depthwise separable convolutions applied independently along the frequency and time axes. Unlike Cai et al. (2024), who explicitly divide the channels into two groups, our approach first reduces the channel dimensionality to $C/2$ through pointwise convolutions, thereby achieving efficient feature mixing without manual channel partitioning.

Given an input feature map $\mathbf{F} \in \mathbb{R}^{B\times C\times H\times W}$ (batch, channels, frequency, time), TF-Attn begins by extracting axis-specific features. A depthwise convolution with a $k_f \times 1$ kernel models local interactions along the frequency axis, while a symmetric $1 \times k_t$ kernel captures temporal dependencies. These two directional filters produce $\mathbf{U}_F$ and $\mathbf{U}_T$, both in $\mathbb{R}^{B\times C\times H\times W}$.

Each branch is then projected from $C$ to $C/2$ channels using a pointwise convolution, producing compact intermediate representations $\mathbf{V}_F$ and $\mathbf{V}_T$. To extract global context along each axis, we apply global average pooling along frequency for the first branch and along time for the second. This yields low-dimensional descriptors of shapes $B \times C/2 \times H \times 1$ (frequency context) and $B \times C/2 \times 1 \times W$ (time context), which are further transformed by a lightweight pointwise convolution. These descriptors are broadcast along the complementary dimension and added back to their respective feature maps, injecting global spectro–temporal structure into each branch.

The context-enhanced tensors are then concatenated along the channel dimension, producing a merged representation $\mathbf{H}$ in $\mathbb{R}^{B\times C\times H\times W}$ that jointly encodes frequency-aware and time-aware information. Finally, a residual connection adds the input $\mathbf{F}$ to the merged output, yielding the final representation $\mathbf{F}_{\text{out}}$. All convolutions in the block follow a Conv–BN–SiLU pattern.

A full mathematical formulation of each operation, including intermediate tensors $\mathbf{U}_F$, $\mathbf{U}_T$, $\mathbf{V}_F$, $\mathbf{V}_T$, $\mathbf{C}_F$, $\mathbf{C}_T$, and the final merge producing $\mathbf{H}$ and $\mathbf{F}_{\text{out}}$, is provided in Appendix C. TF-Attn is inserted throughout our architecture: before each spatial-resolution change in the branch backbones, after feature-fusion layers, and in the top–down pathway of the neck, ensuring that spectro–temporal cues are reinforced across all stages of the network. A schematic illustration of the module is provided in Figure 1.

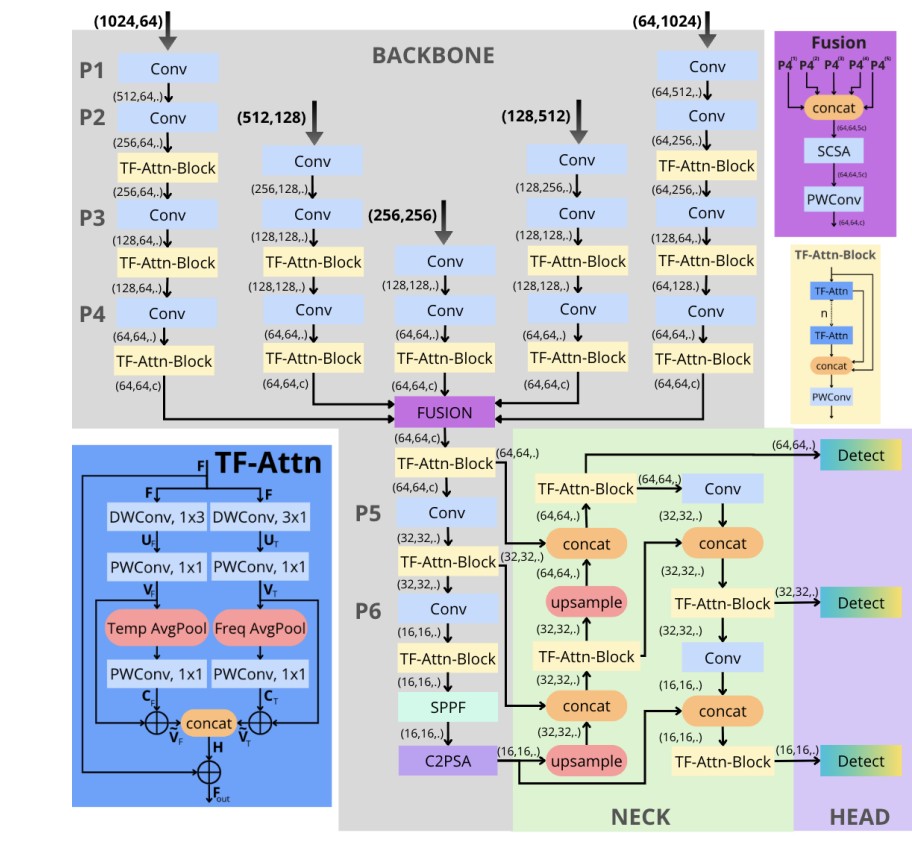

**Figure 1:** TF-Attn-YOLO overall architecture.

## 3.4 DATASET AND EXPERIMENTAL SETUP

We evaluate the proposed architecture on three datasets designed to probe complementary aspects of the problem: (i) a realistic, LPI-rich electronic-warfare benchmark (Dataset A), (ii) an open, LPI-free variant for reproducibility (Dataset B), and (iii) a synthetic, controlled testbed that isolates the benefits of multi-resolution fusion (Dataset C).

Across all datasets, the observed discrete-time baseband signal is modeled as

$$x(t) = \sum_{i=1}^{N} s_i(t) + \eta(t), \tag{2}$$

where each $s_i(t)$ denotes an individual emission and $\eta(t)$ is additive white Gaussian noise (AWGN).

For every acquisition, we compute five spectrograms at distinct STFT scales, using rectangular windows of lengths $L \in \{128, 256, 512, 1024, 2048\}$ samples without overlap. These yield complementary time–frequency resolutions:

$$1024 \times 64, \ 512 \times 128, \ 256 \times 256, \ 128 \times 512, \ 64 \times 1024.$$

All datasets presented below share this multi-resolution representation.

In Datasets A and B, $x(t)$ represents a passively intercepted wideband RF scene in which heterogeneous emitters coexist within the same spectral band. From the interceptor's perspective, this environment introduces two major difficulties: (i) *spectral congestion*, where multiple emissions overlap in the time–frequency plane, and (ii) *blind interception*, where no prior knowledge of the transmitted waveforms is available, ruling out coherent matched filtering and motivating waveform-agnostic, non-coherent detection strategies.

Both datasets contain 100,000 simulated multi-emitter scenarios split into training, validation, and test sets (80%, 10%, 10%). Each scenario corresponds to a full-band RF acquisition of duration

$T = 32.768\,\mu\text{s}$, sampled at $F_s = 4\,\text{GS/s}$, resulting in a Nyquist bandwidth of $B = 2\,\text{GHz}$, and includes up to 10 simultaneously active emitters.

**Dataset A: Full RF interception scenario with LPI emissions.** Dataset A emulates a realistic electronic-warfare environment in which several emitters deliberately seek to evade interception. Low-probability-of-intercept (LPI) radars and other noise-like emissions are engineered to operate at very low SNR and to blend into background interference, thereby defeating classical detection and classification techniques Pace (2009). This makes Dataset A a strong benchmark for assessing the practical relevance of our model in a realistic interception setting.

The dataset includes LFM/NLFM chirps, polyphase codes (P1–P4, Frank), polytime codes (T1–T4), FSK, DSSS-like bursts, unmodulated pulses, and noise-radar LPI waveforms. SNR values are uniformly sampled in $[-15, 15]\,\text{dB}$, with many pulses lying in a regime where individual returns are visually indistinguishable from noise. Due to the sensitive nature of LPI waveform definitions and parameters, this dataset cannot be released publicly; however, it faithfully reflects operational EW interception conditions and constitutes our primary evaluation benchmark.

**Dataset B: Open, LPI-free RF/radar and telecom dataset.** To ensure full reproducibility while maintaining the same blind interception and spectral-congestion conditions as Dataset A, we construct a second dataset using an identical simulation protocol but *excluding* all LPI waveforms. These are replaced by a broader set of communication-like modulations (additional FSK patterns, DSSS-like bursts, OFDM), producing a diverse non-LPI environment with overlapping emissions and variable SNR. The full simulator code needed to regenerate this dataset is publicly released[2], and Dataset B is obtained exactly using `seed=444`.

**Dataset C: Multi-resolution frequency-code disambiguation.** To test whether our architecture can exploit complementary multi-resolution information independently of RF-specific waveform structures, we build a third dataset composed of four synthetic FSK-based codes. Two spectrogram resolutions are used, with STFT window lengths $L = 128$ and $L = 2048$. Each code is constructed from $K$ symbols of duration $T_{\text{sym}} = 128/F_s$, hopping between two tones around a carrier $f_c$. Codes FSK_CODE1 and FSK_CODE2 use a narrow spacing ($\pm\Delta f$, $\Delta f = B/2048$), whereas FSK_CODE3 and FSK_CODE4 use a wider spacing ($\pm 1.5\Delta f$), resolvable only at $L = 2048$. Amplitude alternation distinguishes $\{1, 3\}$ from $\{2, 4\}$, a structure most visible at $L = 128$. By design, no single resolution reveals all discriminative cues, making this dataset a controlled benchmark for evaluating multi-resolution fusion. The exact generation script is included in the public repository[3], and Dataset C is obtained using `seed=974`.

**Training setup and implementation details.** All models (single-resolution baselines and MRS-YOLO) are trained under the same protocol to ensure fair comparison. We use Adam with an initial learning rate of $10^{-3}$, cosine learning-rate decay, and a batch size of 64. Training runs for up to 200 epochs with early stopping based on validation loss. Mixed-precision (AMP) is enabled throughout. Bounding-box classification and regression follows the YOLOv11 formulation with Distribution Focal Loss (`reg_max=16`).

All spectrograms are standardized to zero mean and unit variance per resolution. A fixed train/validation/test split (80/10/10) is used for every dataset, and the entire pipeline (data generation, augmentations, and parameter initialization) is executed with a fixed random seed to ensure determinism. Unless otherwise specified, detections are filtered using standard Non-Maximum Suppression (NMS) with IoU threshold $= 0.5$. For precision–recall analysis, confidence thresholds are selected on the validation set to guarantee 99% precision, as detailed in Section 3.5.

All experiments are implemented in PyTorch and executed on a workstation equipped with $2\times$ NVIDIA H100 NVL GPUs (96 GB each) and a single NVIDIA Tesla V100 GPU (32 GB), driven by dual Intel Xeon Gold 6144 CPUs. No hardware-specific accelerators (TensorRT, custom CUDA kernels, or Triton kernels) are used.

---

[2] `https://github.com/ICLRanonymous2026/ICLR2026DataSimulator/blob/main/examples/rf_data_generation.py`

[3] `https://github.com/ICLRanonymous2026/ICLR2026DataSimulator/blob/main/examples/fsk_codes_generation.py`

## 3.5 RESULTS

Before presenting results, we first define the baselines. As single-resolution baselines, we train five independent **YOLOv11** models, each restricted to a single STFT resolution and matched in size to MRS-YOLO ($\approx$ 2.3M parameters). These baselines represent the strongest possible use of each resolution in isolation and provide the natural point of comparison for evaluating the benefits of explicit multi-resolution fusion.

We also report a theoretical reference, denoted the *oracle-OR*, which acts as an ideal selector that, across the five single-resolution YOLOv11 outputs, keeps only correctly labeled detections. Formally, for each ground-truth instance $g$, let $\mathcal{P}_r(g)$ be the set of predictions at resolution $r$. A detection is declared if: (i) **Detection event:** there exists at least one prediction $p \in \bigcup_r \mathcal{P}_r(g)$ with $\mathrm{IoU}(p, g) \geq 0.5$; (ii) **Class selection:** if multiple predictions satisfy this, only those with the correct class label are retained; (iii) **Box selection:** among the remaining predictions, the one with the highest IoU with $g$ is kept. This oracle thus corresponds to the *best-case combination of independent single-resolution detectors*[4].

**Detection performance.** For a fair comparison, each method's score threshold is tuned to reach 99% precision on the validation set, and recall is plotted versus SNR. A prediction is a true positive if it matches a ground-truth (GT) box with $\mathrm{IoU} \geq 0.5$; otherwise it is a false positive. Figure **??** shows recall vs. SNR at fixed precision $= 99\%$. MRS-YOLO consistently outperforms all individual single-resolution baselines and, importantly, surpasses even the *hypothetical best (oracle-OR)* in the challenging low-SNR regime (from $-15$ to $5\,\mathrm{dB}$) where LPI signals operate. On dataset B, we observe the same trend: the multi-resolution model dominates all single-resolution variants, particularly at low SNR.

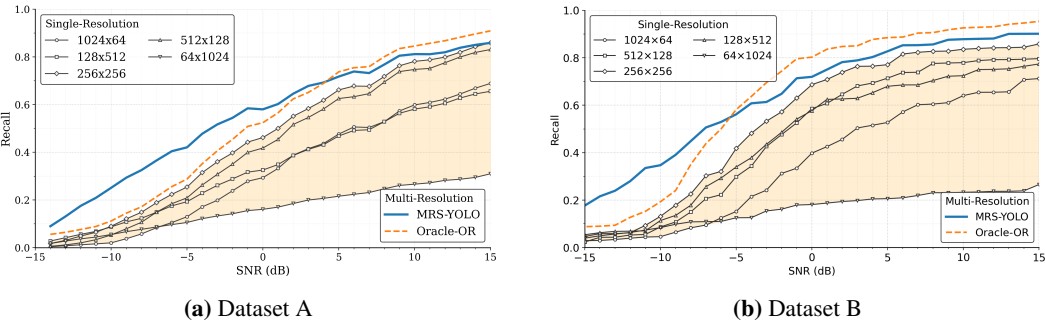

(a) Dataset A                         (b) Dataset B

**Figure 2:** Recall vs. SNR at fixed precision $= 99\%$ (IoU $\geq 0.5$). Grey curves: **single-resolution** YOLOv11 models. Blue solid curve: **MRS-YOLO**. Orange dashed: **hypothetical best (oracle-OR)**.

**Classification performance.** We compare class-conditional predictions using *row-normalized* confusion matrices (Figure 3). Let $\mathbf{R}_A$ and $\mathbf{R}_B$ denote the overall relative confusions for MRS-YOLO and the Oracle-OR, respectively, and let $K$ be the total number of classes. We summarize the improvement with the mean diagonal difference $\Delta_{\mathrm{diag}} = \frac{1}{K} \sum_i \left( [\mathbf{R}_A]_{ii} - [\mathbf{R}_B]_{ii} \right)$. On dataset A, MRS-YOLO achieves $\Delta_{\mathrm{diag}} = +0.055$ (average $+5.5\%$ per class), indicating sharper and more reliable class assignments overall. The gains are particularly strong for **class 3 (frank)** and **class 6 (P3)**, with diagonal improvements of $+39\%$ and $+61\%$, respectively.

On dataset B, single-resolution models frequently confuse QAM telecommunication waveforms with random biphasic phase-transition impulses. MRS-YOLO largely removes this confusion, yielding a **+64.2%** diagonal gain for this waveform and a **+22%** average gain across all classes.

On dataset C (right panel of Figure 3), the 128-sample model mainly confuses **codes 1 and 3** and **codes 2 and 4**, while the 2048-sample model tends to confuse **codes 1 and 2** and **codes 3 and 4**. In contrast, MRS-YOLO clearly separates all four codes, with concentrated diagonals and reduced off-diagonal mass. This shows that the multi-resolution network does more than selecting

---

[4]It is not a general upper bound on multi-resolution learning.

the best single-resolution prediction (as an Oracle would): it effectively combines complementary information across resolutions to disambiguate classes.

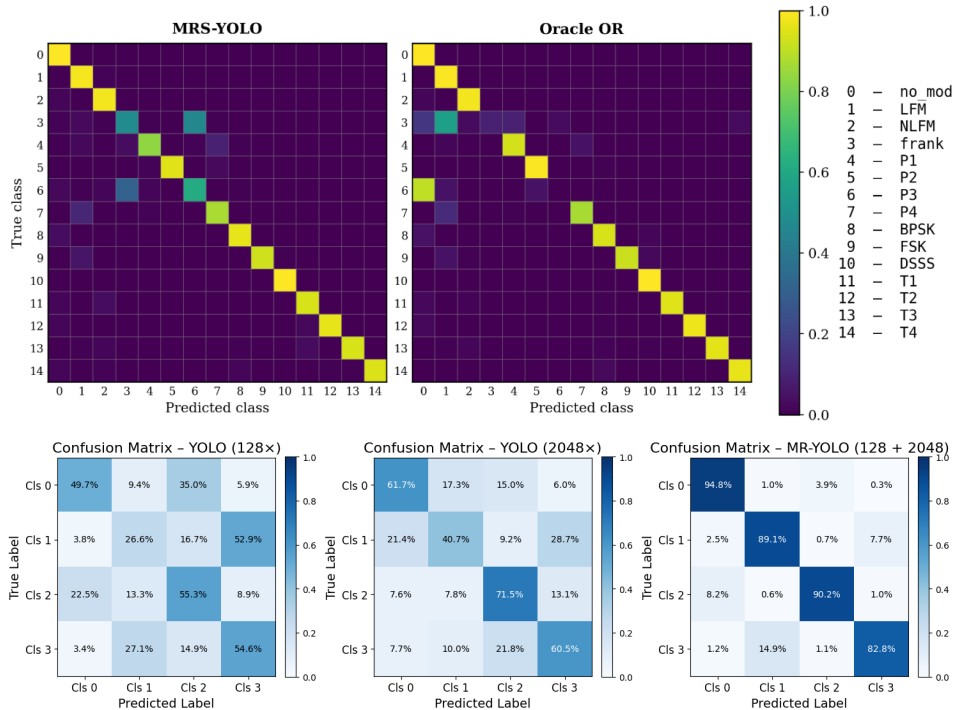

**Figure 3:** Overall row-normalized confusion: **MRS-YOLO** vs the **Oracle-OR** and single-resolution baselines on dataset A (top) and dataset C (bottom).

Tab. 1 and Tab. 2 respectively for Dataset A and B provides a quantitative summary across metrics including mAP and recall at different SNR thresholds. The results confirm that MRS-YOLO achieves the best overall mAP (0.54 / 0.44) on dataset A and (0.60/0.49) for dataset B.

**Table 1:** Overall results with mean average precision and recall at different SNR thresholds for dataset A.

| Model | Params (M) | GFLOPs | mAP50 | mAP50:95 | $\text{Recall}_{\text{SNR} \geq -10\,\text{dB}}$ | $\text{Recall}_{\text{SNR} \geq 0\,\text{dB}}$ | $\text{Recall}_{\text{SNR} \geq 10\,\text{dB}}$ |
|---|---|---|---|---|---|---|---|
| Oracle-OR | $5 \times 2.38$ | $5 \times 1.71$ | 0.40 | 0.37 | 0.58 | **0.76** | **0.88** |
| 1024x64 | 2.38 | 1.71 | 0.14 | 0.11 | 0.37 | 0.52 | 0.64 |
| 512x128 | 2.38 | 1.71 | 0.25 | 0.21 | 0.49 | 0.66 | 0.79 |
| 256x256 | 2.38 | 1.71 | 0.32 | 0.27 | 0.53 | 0.69 | 0.82 |
| 128x512 | 2.38 | 1.71 | 0.30 | 0.25 | 0.39 | 0.51 | 0.62 |
| 64x1024 | 2.38 | 1.71 | 0.20 | 0.15 | 0.19 | 0.24 | 0.29 |
| **MRS-YOLO** | 2.29 | 2.77 | **0.54** | **0.44** | **0.62** | 0.75 | 0.83 |

**Table 2:** Updated detection metrics with recalculated recall values from synthetic SNR curves.

| Model | Params (M) | GFLOPs | mAP50 | mAP50:95 | $\text{Recall}_{\text{SNR} \geq -10\,\text{dB}}$ | $\text{Recall}_{\text{SNR} \geq 0\,\text{dB}}$ | $\text{Recall}_{\text{SNR} \geq 10\,\text{dB}}$ |
|---|---|---|---|---|---|---|---|
| Oracle-OR | $5 \times 2.38$ | $5 \times 1.71$ | 0.48 | 0.43 | **0.755** | **0.908** | **0.938** |
| 1024x64 | 2.38 | 1.71 | 0.19 | 0.15 | 0.467 | 0.610 | 0.694 |
| 512x128 | 2.38 | 1.71 | 0.32 | 0.27 | 0.599 | 0.749 | 0.800 |
| 256x256 | 2.38 | 1.71 | 0.38 | 0.32 | 0.674 | 0.813 | 0.855 |
| 128x512 | 2.38 | 1.71 | 0.36 | 0.30 | 0.590 | 0.718 | 0.775 |
| 64x1024 | 2.38 | 1.71 | 0.23 | 0.18 | 0.184 | 0.220 | 0.245 |
| **MRS-YOLO** | 2.29 | 2.77 | **0.60** | **0.49** | **0.750** | 0.853 | 0.894 |

## 3.6 ABLATION STUDY

To disentangle the contribution of each design choice in MRS-YOLO, we perform a structured ablation study in three stages. We begin by establishing a strong single-resolution baseline as our reference. We then explore different strategies for extending this baseline to multi-resolution processing

through alternative backbone architectures. Finally, we refine the selected design by evaluating the impact of attention-based fusion modules and specialized time–frequency attention blocks.

**Single-Resolution Baselines.** We begin by evaluating the performance of state-of-the-art one-stage object detectors adapted to spectrogram inputs, each trained on a single-resolution STFT. This step allows us to select a fair reference for subsequent comparisons.

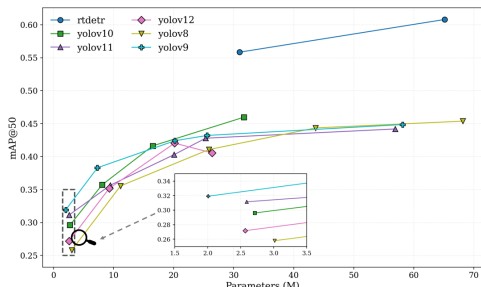

At higher computational budgets, RT-DETR Zhao et al. (2024) exhibits a clear performance gap over the YOLO variants. Conversely, in the lightweight regime ($< 5M$ parameters), *YOLOv11* and *YOLOv9* stand out. We adopt *YOLOv11* as our reference baseline: it follows a more conventional architecture, whereas *YOLOv9* relies on a training–inference discrepancy, complicating fair comparisons.

**Figure 4:** mAP@50 vs. Params for YOLO (v8–v12) and RT-DETR variants trained on spectrum-$256 \times 256$.

In summary, we retain the main architectural components of YOLOv11, namely the `C3k2` blocks, the `SPPF+C2PSA` module and its `Detect` blocks, as the building blocks from which we construct our multi-resolution architecture.

**Multi-Resolution Backbone Designs.** We next investigate how to extend YOLOv11 to process multiple spectrogram resolutions. We compare three variants. In the **Multi-Fusion design**, each resolution is processed by its own backbone up to feature maps $P_4$–$P_6$, which are then fused across resolutions before entering the neck. This maximizes per-resolution capacity but involves redundant computation across branches. In the **Pyramidal Downsampling backbone** (our final choice), each branch produces a resolution-specific $P_4^{(r)}$; these are fused once at the $P_4$ level, and the fused map is shared across deeper stages to generate $P_5$ and $P_6$, thus avoiding repeated processing after fusion. Finally, in the **Max-Resolution Upsampling backbone**, all inputs are upsampled to the largest resolution, concatenated channel-wise, and processed by a single YOLO backbone, which is structurally inefficient since the network always operates on the largest tensors.

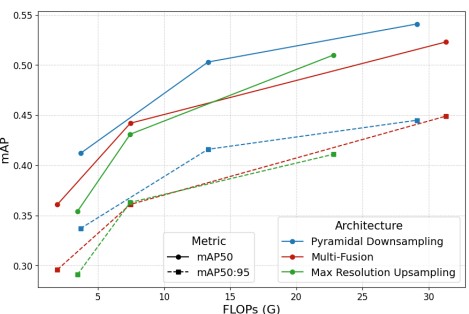

Figure 5 reports mAP@50 (solid) and mAP@50:95 (dashed) as functions of FLOPs. Across all computational budgets, the **pyramidal design consistently outperforms the alternatives**, demonstrating a better accuracy–efficiency trade-off. While Multi-Fusion and Max-Res Upsampling can reach competitive performance, they are structurally less efficient since they either replicate computation across branches or operate entirely at the largest resolution. We therefore adopt the pyramidal backbone as the default architecture of MRS-YOLO.

**Figure 5:** Comparison of multi-resolution backbone designs. Model capacity is controlled by varying the number of channels per layer, so that different designs can be compared at similar FLOPs.

**Fusion Strategy.** Having fixed the backbone, the next key design choice concerns how multi-resolution features are fused. A naive concatenation of all branches retains the full information but also introduces redundancy, leading to increased computational cost and suboptimal filtering when followed only by a pointwise convolution. To address this, we benchmark several attention modules from the literature, inserted between concatenation and channel reduction, which aim to better highlight informative spectro-temporal patterns while suppressing less relevant or redundant

information. We evaluate: **CA** (channel attention), **SA** (spatial attention), **CBAM** (CA+SA (Woo et al., 2018)), **PCSA**, **SCAM**, **SCSA** (Si et al., 2025), and **A2C2f** area attention (Tian et al., 2025). Among the tested options, *SCSA* offers the most favorable balance between accuracy and efficiency, improving both mAP@50 and mAP@50:95 by 6 points over the no-attention baseline. We therefore adopt it as the default fusion module in MRS-YOLO.

**TF-Attn Block.** Conventional `C3k2` and `C2f` blocks, inherited respectively from Yolov8 (Jocher et al., 2023) and Yolov11 (Jocher & Qiu, 2024), are not tailored to capture the structured patterns of spectrograms. As shown in Table 3 (C), replacing them with lightweight TF-attention modules consistently improves detection. Our TF-Attn block provides the best overall compromise: with only 2.29M parameters (22% fewer) and 2.77 GFLOPs (20% fewer), it achieves the highest accuracy, reaching mAP@50 = **0.54** and mAP@50:95 = **0.44**.

**Unified Ablation Results.** Table 3 consolidates all results, covering (A) backbone architectures, (B) fusion strategies, and (C) block variants. This unified view highlights three key findings: (1) pyramidal downsampling is the most efficient multi-resolution backbone; (2) attention-based fusion consistently improves accuracy, with SCSA performing best; (3) our TF-Attn block provides the strongest gains while remaining lightweight. These choices define the final MRS-YOLO design.

**Table 3:** Ablation study of multi-resolution backbone, fusion strategies, and block designs in MRS-YOLO. Best values are in bold; the final configuration (Pyramidal + SCSA + TF-Attn) is highlighted.

| Backbone | Fusion | Block | Params (M) | FLOPs (G) | mAP@50 | mAP@50:95 |
|---|---|---|---|---|---|---|
| **(A) Backbone architectures** | | | | | | |
| Pyramidal | None | C3k2 Jocher & Qiu (2024) | 2.90 | 3.39 | **0.41** | **0.33** |
| Multi-Fusion | None | C3k2 | 3.56 | 1.94 | 0.36 | 0.30 |
| Max-Res Upsampling | – | C3k2 | 2.73 | 3.45 | 0.35 | 0.29 |
| **(B) Attention in Fusion (Backbone = Pyramidal, Block = C3k2)** | | | | | | |
| Pyramidal | CA | C3k2 | 2.94 | 3.48 | 0.46 | 0.37 |
| Pyramidal | SA | C3k2 | 2.93 | 3.48 | 0.49 | 0.40 |
| Pyramidal | CBAM Woo et al. (2018) | C3k2 | 2.94 | 3.48 | 0.47 | 0.39 |
| Pyramidal | PCSA | C3k2 | 2.93 | 3.48 | 0.46 | 0.37 |
| Pyramidal | SCAM | C3k2 | 2.94 | 3.48 | 0.50 | 0.41 |
| Pyramidal | SCSA Si et al. (2025) | C3k2 | 2.94 | 3.48 | **0.52** | **0.43** |
| Pyramidal | A2C2f Tian et al. (2025) | C3k2 | 3.03 | 3.91 | 0.47 | 0.39 |
| **(C) Block variants (Backbone = Pyramidal, Fusion include SCSA)** | | | | | | |
| Pyramidal | SCSA | C2f Jocher et al. (2023) | 3.12 | 3.97 | 0.46 | 0.37 |
| Pyramidal | SCSA | C3k2 | 2.94 | 3.48 | 0.52 | 0.43 |
| Pyramidal | SCSA | TFA Zha et al. (2019) | 1.94 | 2.23 | 0.36 | 0.28 |
| Pyramidal | SCSA | TFA Lin et al. (2022) | 2.38 | 2.91 | 0.51 | 0.42 |
| Pyramidal | SCSA | TFSepConv Cai et al. (2024) | 2.27 | 2.69 | 0.53 | 0.42 |
| Pyramidal | SCSA | TF-Attn (ours) | 2.29 | 2.77 | **0.54** | **0.44** |

# 4 CONCLUSION

We introduced MRS-YOLO, a lightweight multi-resolution extension of YOLO tailored to time–frequency signal detection. The model processes spectrograms at complementary resolutions through parallel branches and fuses them with an attention-based mechanism, while a dedicated time–frequency attention block jointly captures temporal and spectral patterns. Experiments on a large-scale synthetic RF dataset show consistent gains over strong single-resolution baselines: at fixed false-alarm rates, MRS-YOLO improves recall across all SNRs and enhances class discrimination in dense spectral conditions. These findings underscore the value of explicit multi-resolution fusion in compact architectures and establish MRS-YOLO as a new benchmark for TF signal detection.

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

APPENDIX

## A  TESTED BACKBONE ARCHITECTURES

In Figure 6, we illustrate the three multi-resolution backbone designs evaluated in this work.

## B  FULL MODEL SPECIFICATION

Table 4 provides the full specification of the MRS-YOLO architecture used in Section 3.5

**Table 4: Architecture summary of MRS-YOLO.** Pyramidal multi-resolution backbone (five branches), $P_4$ fusion via SCSA/PCSA, neck across $P_4$–$P_6$, SPPF + C2PSA, PAN-like neck, and three-scale detection with DFL. Strides are reported per stage; effective output strides follow YOLO conventions ($\sim 8/16/32$ relative to the input).

| Section | Submodule | $C_{in} \to C_{out}$ | Stride | Blocks / Details | Role |
|---|---|---|---|---|---|
| Backbone | 5 branches | – | – | Stem convs + TF-Attn-Blocks (n=1) | Multi-resolution encoding |
| Branch 0 | Conv(1→32)→Conv(32→64) | 1→32→64 | (2,2),(2,2) | 2×TF-Attn-Block, then 1×TF-Attn-Block | To $P_4$ |
| Branch 1 | Conv(1→16)→···→64 | 1→16→16→32→64 | (1,2)×4 | 4×TF-Attn-Block | To $P_4$ |
| Branch 2 | Conv(1→16)→···→64 | 1→16→16→32→64 | (2,1)×4 | 2×TF-Attn-Block | To $P_4$ |
| Branch 3 | Conv(1→16)→32→64 | 1→16→32→64 | (2,1),(2,1),(2,2) | 2×TF-Attn-Block | To $P_4$ |
| Branch 4 | Conv(1→16)→32→64 | 1→16→32→64 | (1,2),(1,2),(2,2) | 2×TF-Attn-Block | To $P_4$ |
| $P_4$ fusion | fuse_p4 | 320→64 | 1 | SCSA = SMSA(3/5/7/9) + PCSA; Conv 1×1 | Alignment/weighting at $P_4$ |
| $P_4$ neck | c4_p4 | 64→64 | 1 | TF-Attn-Block | $P_4$ refinement |
| $P_5$ down | conv_p5 | 64→128 | 2 | Conv 3×3 | Downsample to $P_5$ |
| $P_5$ neck | c4_p5 | 128→128 | 1 | TF-Attn-Block | $P_5$ refinement |
| $P_6$ down | conv_p6 | 128→256 | 2 | Conv 3×3 | Downsample to $P_6$ |
| $P_6$ neck | c4_p6 | 256→256 | 1 | TF-Attn-Block | $P_6$ refinement |
| Bottleneck | sppf | 256→256 | 1 | SPPF (MaxPool 5, concat) | Context aggregation |
| Attention | psa (C2PSA) | 256→256 | 1 | 2×Conv 1×1 | Channel/spatial selection |
| Head | upsample | – | ×2 | Nearest | Up: $P_6 \to P_5$, $P_5 \to P_4$ |
| Head $P_6 \to P_5$ | head_c4_1 | 384→128 | 1 | Conv 1×1 + TF-Attn-Block | Top–down fusion |
| Head $P_5 \to P_4$ | head_c4_2 | 192→64 | 1 | Conv 1×1 + TF-Attn-Block | Top–down fusion |
| Down $P_4$ | down_p4 | 64→64 | 2 | Conv 3×3 | $P_4 \to P_5$ (PAN) |
| Head $P_4 \to P_5$ | head_c4_3 | 192→128 | 1 | Conv 1×1 + TF-Attn-Block | Bottom–up fusion |
| Down $P_5$ | down_p5 | 128→128 | 2 | Conv 3×3 | $P_5 \to P_6$ (PAN) |
| Head $P_5 \to P_6$ | head_c4_4 | 384→256 | 1 | Conv 1×1 + TF-Attn-Block | Bottom–up fusion |
| Detect | dist branches (×3) | . →64 | 1 | {Conv 3×3, Conv 3×3, Conv 1×1→64} | Box distribution |
| | cls branches (×3) | . →**15** | 1 | {Conv 3×3, Conv 3×3, Conv 1×1→15} | 15 classes |

## C  DETAILED FORMULATION OF TF-ATTN

For completeness, we provide here the mathematical formulation of the TF-Attn block. Given an input feature map $\mathbf{F} \in \mathbb{R}^{B \times C \times H \times W}$, the block begins with two directional depthwise convolutions:

$$\mathbf{U}_F = \mathrm{DW}_{k_f \times 1}(\mathbf{F}) \quad \in \mathbb{R}^{B \times C \times H \times W} \quad \text{(frequency filtering)},$$

$$\mathbf{U}_T = \mathrm{DW}_{1 \times k_t}(\mathbf{F}) \quad \in \mathbb{R}^{B \times C \times H \times W} \quad \text{(temporal filtering)}.$$

Each branch is projected to $C/2$ channels:

$$\mathbf{V}_F = \mathrm{PW}_{C \to C/2}(\mathbf{U}_F) \quad \in \mathbb{R}^{B \times C/2 \times H \times W} \quad \text{(frequency branch)},$$

$$\mathbf{V}_T = \mathrm{PW}_{C \to C/2}(\mathbf{U}_T) \quad \in \mathbb{R}^{B \times C/2 \times H \times W} \quad \text{(time branch)}.$$

Axis-specific context descriptors are obtained through global average pooling and a pointwise projection:

$$\mathbf{C}_F = \mathrm{PW}(\mathrm{GAP}_F(\mathbf{V}_F)) \quad \in \mathbb{R}^{B \times C/2 \times H \times 1} \quad \text{(frequency context)},$$

$$\mathbf{C}_T = \mathrm{PW}(\mathrm{GAP}_T(\mathbf{V}_T)) \quad \in \mathbb{R}^{B \times C/2 \times 1 \times W} \quad \text{(time context)}.$$

The contexts are broadcast along the complementary axis and added to their respective branches:

$$\tilde{\mathbf{V}}_F = \mathbf{V}_F \oplus \mathbf{C}_F, \qquad \tilde{\mathbf{V}}_T = \mathbf{V}_T \oplus \mathbf{C}_T.$$

Finally, the two branches are concatenated along channels and combined with a residual connection:

$$\mathbf{H} = \mathrm{Concat}_c(\tilde{\mathbf{V}}_F, \tilde{\mathbf{V}}_T) \in \mathbb{R}^{B \times C \times H \times W}, \qquad \mathbf{F}_{\mathrm{out}} = \mathbf{H} + \mathbf{F}.$$

All convolutions follow a Conv–BN–SiLU pattern, and $\oplus$ denotes broadcast addition along singleton dimensions.

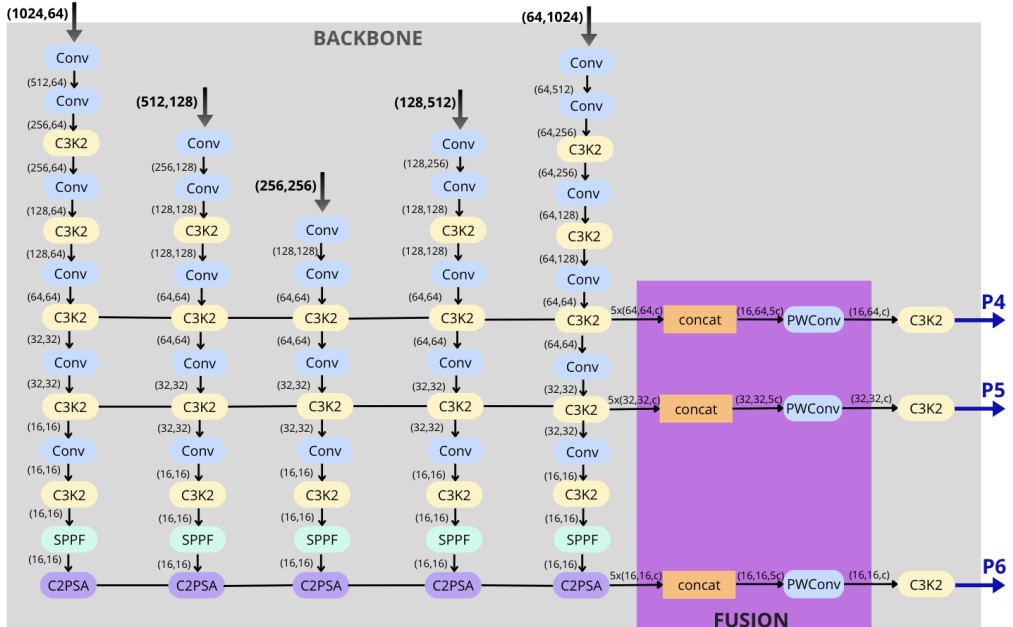

(a) Multi-Fusion design: backbones up to $P_6$, followed by fusion at $P_4$, $P_5$ and $P_6$.

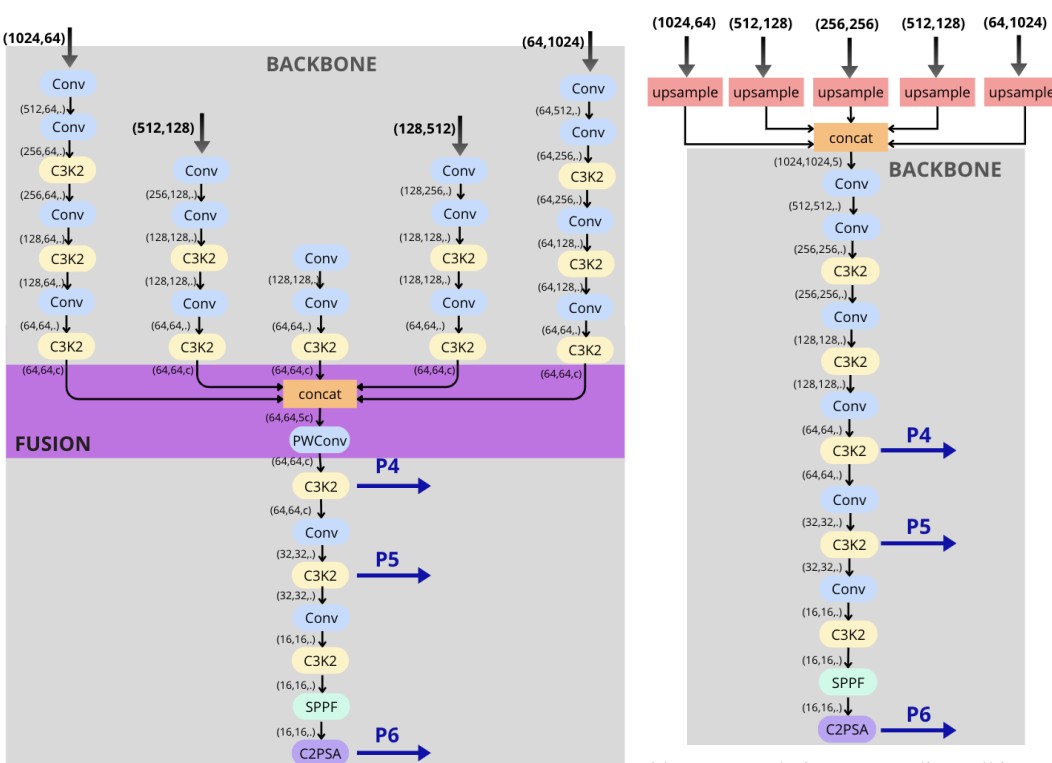

(b) Pyramidal Downsampling: fusion occurs once at $P_4$, then shared downsampling produces $P_5$ and $P_6$.

(c) Max-Resolution Upsampling: all inputs are resized to the largest scale and concatenated before a single backbone.

**Figure 6:** Comparison of backbone designs. (a) Multi-Fusion with full per-resolution branch; (b) Pyramidal Downsampling (final choice), sharing computation beyond $P_4$; (c) Max-Resolution Upsampling with a single backbone at the largest scale.

# D INFERENCE BENCHMARKS

We report the per-image inference time of both the single-resolution YOLO models and the full multi-resolution MRS-YOLO model. All five operational STFT resolutions used in the main experiments $(1024 \times 64), (512 \times 128), (256 \times 256), (128 \times 512), (64 \times 1024)$ exhibit almost identical latency on all hardware configurations, with differences. For clarity, we therefore report only the representative case $(256 \times 256)$ for the single-resolution YOLO baseline. We also include larger square resolutions to illustrate how latency scales when input size increases. Finally, we report the latency of the full MRS-YOLO model, which processes the five resolutions jointly in a single forward pass. All measurements were obtained with batch size 64, 10 warm-up iterations, explicit device synchronization before and after each forward pass, and timings averaged over 100 iterations on GPU (50 on CPU).

In addition, it is important to note that the time–frequency attention layers, although residual in terms of parameter count, introduce a slight increase in computation time. Since these blocks improve performance, it becomes possible to reduce overall inference time while maintaining accuracy by decreasing the number of parameters compared with a model without attention. It must also be considered that the baselines reported here rely on isotropic single-resolution models, performing the same amount of downsampling on the time and frequency axes even when their dimensions differ. Anisotropic single-resolution models, constructed in the same spirit as MRS but without the multi-resolution design, would require deeper downsampling on the larger axis and therefore yield higher inference times.

**Table 5:** Per-image inference time (ms) for single-resolution YOLO (representative and large resolutions) and full multi-resolution MRS-YOLO.

| Model / Resolution | H100 (ms) | V100 (ms) | CPU (ms) |
|---|---|---|---|
| YOLO (256×256) | $0.112 \pm 0.003$ | $0.195 \pm 0.005$ | $6.762 \pm 0.071$ |
| YOLO (512×512) | $0.198 \pm 0.008$ | $0.674 \pm 0.007$ | $38.031 \pm 0.727$ |
| YOLO (1024×1024) | $0.824 \pm 0.010$ | $2.764 \pm 0.004$ | $180.738 \pm 2.045$ |
| **MRS-YOLO (5 resolutions)** | $\mathbf{0.792 \pm 0.009}$ | $\mathbf{2.596 \pm 0.005}$ | $\mathbf{151.533 \pm 5.751}$ |

# E DISCUSSION ON THE USE OF SIMULATED DATASETS

This work relies on simulated datasets for training and evaluating MRS-YOLO. We explain here why this choice is natural in our target application domain, why it is difficult to replace with publicly available real data, and how it affects the interpretation of our results.

The primary motivation comes from *electronic warfare (EW)* and RF spectrum monitoring. In this context, large-scale RF datasets with dense time–frequency annotations (event start and end times, carrier frequency, bandwidth, and class labels) are rarely accessible: recordings are typically sensitive, and producing detailed annotations is technically expensive and almost never done at scale. As a consequence, it is standard practice in EW to develop and validate detection models using simulated or semi-simulated signals that approximate operational conditions (waveforms, SNR ranges, interference, clutter), and then deploy these models on real data. Our use of simulated datasets is therefore fully consistent with established practice in *electronic warfare* applications.

Beyond EW-specific scenarios, we did not evaluate on real open-source RF or acoustic datasets for a simple reason: reproducing our experimental pipeline requires a dataset that simultaneously provides (i) raw 1D signals to compute multiple STFTs at different window lengths, (ii) dense time–frequency annotations in the form of bounding boxes and classes, (iii) enough data to train and evaluate deep learning models, and (iv) sufficiently challenging scenes (multiple simultaneous signals, low SNR, overlapping events) to justify the use of a multi-resolution detector. To the best of our knowledge, no public dataset satisfies all these constraints. Existing RF corpora typically provide clip-level labels (e.g., modulation type) without time–frequency bounding boxes, while most audio or bioacoustic datasets provide partial annotations (timestamps or weak labels) but not the combination of raw 1D data, multi-resolution TF access, and dense bounding boxes required in our setting.

To compensate for this lack of suitable public data, we rely on a simulator that is detailed and physically grounded. It models interference, overlapping emissions, Doppler effects, realistic rise/fall times, propagation attenuation, and waveform libraries whose characteristics match documented specifications. The acquisition chain is based on an actual operational system. This level of realism ensures that the synthetic data capture the key phenomena encountered in real operational scenarios, thereby preserving the relevance of the evaluation despite the absence of public real-world datasets. We view our simulator and synthetic datasets as a first step toward more systematic benchmarks for multi-resolution time–frequency detection, and we consider evaluation on suitably annotated real-world corpora, once such datasets become available, as an important direction for future work.

## F  QUALITATIVE RESULTS

To complement the quantitative evaluation, we provide qualitative examples illustrating the benefits of multi-resolution processing. For each selected scenario, we display the raw spectrograms at the five STFT resolutions ($1024 \times 64$, $512 \times 128$, $256 \times 256$, $128 \times 512$, $64 \times 1024$), highlighting the complementary visibility of signal events across scales.

In addition, we show detection results on the $256 \times 256$ spectrogram, which serves as a representative mid-resolution view. Ground-truth bounding boxes are drawn in green, while predicted boxes from MRS-YOLO are shown in red.

**Scenario description.** The qualitative example corresponds to a mixed environment with four radar emitters and one interfering telecom source. Each source transmits a distinct waveform type with different bandwidths and signal-to-noise ratios (SNR/INR), producing heterogeneous visibility across STFT resolutions. For clarity of visualization, we intentionally selected a scenario that is not overly congested, so that individual waveforms remain visually distinguishable on the spectrograms. For the same reason, we mainly chose signals with relatively high SNR, ensuring that their structures are clearly visible across resolutions.

- **Emitter 1**: NLFM short waveform, carrier frequency $f_p \approx 0.39$ GHz, bandwidth $\approx 269$ MHz, pulse width $0.58$ $\mu$s, SNR $\approx -5.2$ dB.
- **Emitter 2**: waveform type P4, $f_p \approx 0.88$ GHz, bandwidth $\approx 1.35$ GHz, pulse width $5.3$ $\mu$s, SNR $\approx +6.8$ dB.
- **Emitter 3**: LFM short waveform, $f_p \approx 0.72$ GHz, bandwidth $\approx 696$ MHz, pulse width $0.93$ $\mu$s, SNR $\approx +0.5$ dB.
- **Emitter 4**: unmodulated pulse, $f_p \approx 1.25$ GHz, negligible bandwidth, pulse width $0.48$ $\mu$s, SNR $\approx +13.3$ dB.
- **Interference 1**: telecom DSSS waveform, $f_p \approx 1.04$ GHz, bandwidth $\approx 656$ MHz, duration $32.8$ $\mu$s, interference-to-noise ratio (INR) $\approx +5.2$ dB.

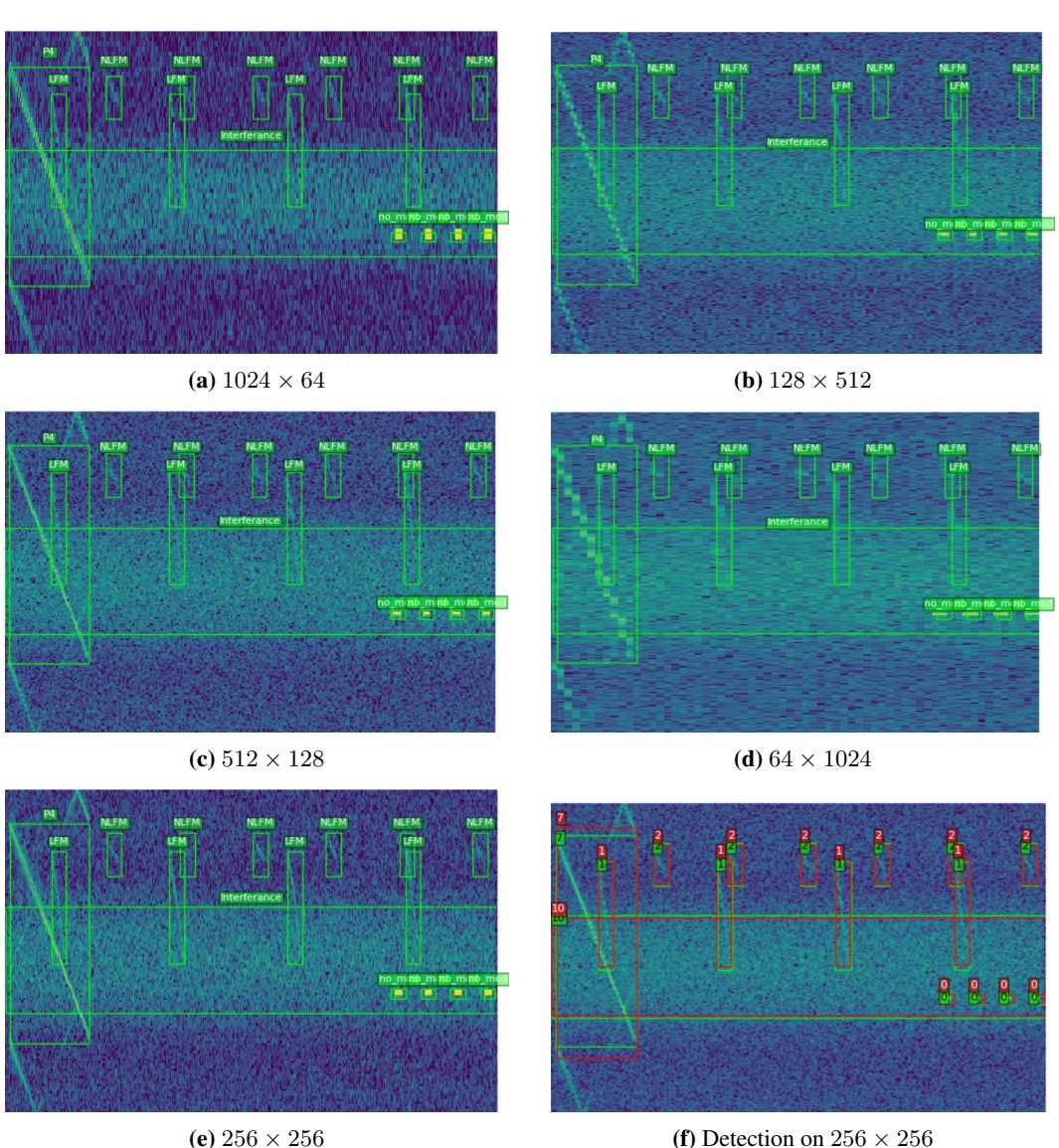

**(a)** $1024 \times 64$      **(b)** $128 \times 512$

**(c)** $512 \times 128$      **(d)** $64 \times 1024$

**(e)** $256 \times 256$      **(f)** Detection on $256 \times 256$

**Figure 7: Qualitative example on dataset A.** Ground-truth boxes are shown in green and predictions in red. In this scenario, MRS-YOLO successfully detects all pulses, illustrating its robustness under the chosen conditions.

