# OpenReview forum: "MRS-YOLO : A YOLO model for signal detection in multi-resolution spectrograms"
_ICLR.cc/2026/Conference — ICLR 2026 Conference Desk Rejected Submission_

### Official Review · Reviewer_p72W · 2025-10-30

**Soundness:** 2
**Presentation:** 3
**Contribution:** 2
**Rating:** 2
**Confidence:** 4

**Summary:**

This paper proposes MRS-Yolo, a Yolo-based model for signal detection in multi-resolution spectrograms. The idea of using multi-resolution analysis is well-motivated in digital signal processing, though similar approaches have been explored in previous works.

**Strengths:**

- In the abstract, the connection between Heisenberg uncertainty and spectrogram resolution is unclear.

- There is extensive prior work on multi-resolution spectrogram feature extraction, but the paper does not adequately cite related references.

**Weaknesses:**

- In the abstract, the connection between Heisenberg uncertainty and spectrogram resolution is unclear.

- There is extensive prior work on multi-resolution spectrogram feature extraction, but the paper does not adequately cite related references.

Weakness

- The main concern is the lack of novelty. Many prior works propose similar solutions; Sections 3.2 and 3.3 do not clearly demonstrate innovative contributions.

- The equations in Section 3.3 are verbose and could be moved to an appendix for clarity.

- The evaluation uses simulated data, which limits confidence in the proposed method’s real-world performance.

- The evaluated dataset consists solely of radio-frequency signals, which may not demonstrate the general significance of the proposed system. For a new architecture, evaluation on diverse datasets from multiple domains is preferred.

- It is unclear whether Yolo requires adaptation for the proposed features. How is the domain gap between image signals and radio signals addressed?

- Comparison with Wavelet Transform is missing. Since wavelets also provide multi-resolution analysis, it would be informative to compare against them or other standard methods to validate the effectiveness of MRS-Yolo.

**Questions:**

N.A.

**Details Of Ethics Concerns:**

N.A.

---

> ### Author Response · Authors · 2025-11-26
> **To Assigned Reviewer p72W**
>
> We thank the reviewer for the constructive and detailed feedback. The remarks have helped us improve the clarity, structure, and positioning of the revised manuscript. Below, we address each point raised.
>
> 1. **Clarifying the connection between Heisenberg uncertainty and spectrogram resolution**
> In the revised manuscript, we have rewritten the abstract to state more clearly how the Heisenberg uncertainty principle motivates the use of multiple spectrogram resolutions.
>
> 2. **Prior work on multi-resolution spectrogram representations**
> The reviewer notes that extensive prior work exists on multi-resolution time–frequency representations. In the Related Work section, we already include a subsection entitled *Multi-Resolution Signal Representations*. However, we acknowledge the reviewer’s concern that the survey could be further strengthened. If the reviewer has specific references in mind, we will gladly integrate them. Absent such references, we respectfully clarify the novelty claim:
>
> - Multi-resolution feature extraction for classification has been explored in prior work.
> - However, to the best of our knowledge there is no prior work addressing full object detection (precise localization + classification) in multi-resolution spectrograms.
>
> This distinction is central: feature-level multi-resolution processing does exist, but multi-resolution TF object detection remains unaddressed in the literature. We have revised the manuscript to clarify this distinction.
>
> 3. **Use of simulated data and real-world generalization**
> In the revised version of the manuscript, we have added a dedicated appendix section: *“Discussion on the Use of Simulated Datasets”*, that directly addresses these questions.
>
> 4. **Evaluation on non-RF datasets**
> We emphasize that we did not identify any publicly available real-world dataset that satisfies the requirements for multi-resolution time–frequency detection: raw 1D signals to compute several STFT resolutions, dense bounding-box annotations in the TF plane, multiple classes, and sufficient data volume for training deep detectors. For this reason, we were unable to include real non-RF benchmarks, and we remain open to any dataset suggestions from the reviewers.
>
> To nevertheless assess cross-domain generalization, the revised manuscript introduces an additional experiment on a fully synthetic frequency-coded dataset. These signals form abstract time–frequency patterns, and several classes are intentionally designed to be indistinguishable at a single resolution. The results show that MRS-YOLO successfully separates these classes using multi-resolution inputs, demonstrating that the benefits of the architecture extend beyond RF-specific scenarios.
>
> 5. **Adaptation of YOLO to spectrogram inputs**
> The reviewer asks how the domain gap between images and spectrograms is handled. We clarify this point explicitly in the revised version:
>
> The architecture must be adapted because spectrograms exhibit strong anisotropy (time and frequency scales differ), unlike isotropic natural images, so we used anisotropique convolutionnal branchs.
>
> Also the time-frequency attention are specificaly pertinent for the time frequency properties of the acquisition and were never employed in a YOLO like model.
>
> This clarification is now emphasized in Sections 3.1 and 3.2.
>
> 6. **Comparison with wavelet transforms**
> We thank the reviewer for raising this point. In the revised version, we clarify why our approach relies on multi-resolution STFTs rather than other TF transforms. Although wavelets provide a form of multi-resolution analysis, their dyadic decimation leads to a frequency–duration trade-off that is not well suited to our setting: in our application, signal duration is not intrinsically tied to its central frequency, making the wavelet hierarchy suboptimal. In addition, wavelet-based representations are more computationally demanding and less directly compatible with existing TF object-detection architectures. In contrast, STFT-based spectrograms offer uniform and interpretable TF grids and integrate naturally with modern detection backbones such as YOLO.

---

### Official Review · Reviewer_vBRR · 2025-10-31

**Soundness:** 3
**Presentation:** 3
**Contribution:** 3
**Rating:** 6
**Confidence:** 3

**Summary:**

This paper proposes MRS-YOLO, a multi-resolution extension of the YOLO object detection framework, designed to identify events in time–frequency spectrograms of complex radio-frequency (RF) signals. The model processes several spectrograms computed at different STFT window lengths through parallel convolutional branches and fuses them with a synergistic attention block (SCSA). A lightweight Time–Frequency Attention (TF-Attn) module reinforces spectral–temporal structure throughout the backbone and neck. The work claims improvements in detection recall and classification accuracy under low-SNR conditions, outperforming both single-resolution YOLO baselines and an oracle-based fusion reference.

**Strengths:**

Strengths
1. The multi-resolution formulation is well-motivated by the Heisenberg uncertainty trade-off, and the analogy to multi-scale vision detection is articulated with clarity and precision.
2.  The integration of parallel spectrogram branches with pyramidal fusion is elegant and computationally efficient. The SCSA-based fusion and TF-Attn block demonstrate an insightful adaptation of attention to spectro-temporal data.
3. The authors conduct a detailed series of ablations isolating the effect of backbone design, fusion strategy, and attention mechanism, giving the reader confidence in the architectural choices.
4.  The paper provides multiple metrics (mAP, recall at varying SNRs, confusion matrices) and well-controlled baselines (five single-resolution YOLOv11 variants plus an oracle ensemble).
5. The structure is methodical, figures and tables are generally informative, and the technical descriptions are mathematically consistent.

**Weaknesses:**

Weaknesses
1. The dataset is fully synthetic and unavailable due to security concerns. Without public access or at least detailed signal specifications, reproducibility and external validation are limited. The lack of evaluation on any real RF or audio dataset weakens claims of generality.
2. All baselines are internal YOLO variants. There is no comparison against other contemporary TF-detection models (e.g. Transformer-based or CNN–RNN hybrids) beyond citation. Including such results would help quantify the practical significance of the proposed method.
3. While numerous design factors are tested, statistical variation (e.g. standard deviation across runs) and significance tests are missing. The improvements, though consistent, may be within normal training noise.
4. The novelty is incremental rather than fundamental—essentially a careful engineering synthesis of known ideas (multi-resolution processing + attention-based fusion). The manuscript could benefit from a more critical discussion of why TF-Attn or SCSA outperform simpler attention blocks in spectrogram domains.
5. Although runtime efficiency is claimed, FLOPs and latency are compared inconsistently across models. The fairness of these comparisons (especially versus the oracle ensemble) should be clarified.
6. A few sections verge on tutorial style; condensing the literature survey would improve focus. The “LLM usage disclosure” is unnecessary for a scientific paper and should be omitted in a final version.
7. No results on transferability, robustness to unseen modulation types, or sensitivity to window parameter choices are provided. These are crucial for applications in electronic warfare or spectrum monitoring.

**Questions:**

plz see my detailed comments above

---

> ### Author Response · Authors · 2025-11-26
>
> We thank the reviewer for the detailed and insightful feedback. We appreciate both the positive assessment of our contributions and the constructive remarks, which have guided several substantial improvements in the revised manuscript.
>
> Below, we address each of the reviewer’s concerns.
>
> **1. Reproducibility and reliance on simulated data**
>
> This was the most significant point raised across all reviews. In the revised version, we now provide a **fully reproducible public dataset** together with the **complete simulation pipeline** (excluding classified electronic-warfare waveforms).
>
> - The dataset includes both telecommunications and conventional radar signals.
> - It is fully annotated for time–frequency object detection and supports the computation of multiple STFT resolutions.
> - We provide pseudo-code and implementation scripts enabling users to regenerate the dataset as well as the entire experimental protocol.
> - We have reproduced all main experiments using this public dataset, and the updated results are now integrated into the revised *Results* section.
>
> To address the broader question of relying on simulated data, we have added a dedicated appendix section (*Discussion on the Use of Simulated Datasets*), which explains the motivation for simulation in electronic-warfare applications, the absence of suitable real-world public datasets, and the limitations and implications of this choice.
>
> This revision resolves the reproducibility concerns while preserving the realism of the low-SNR conditions central to our study.
>
> **2. Lack of baselines beyond YOLO variants**
>
> Our focus on YOLO variants is intentional: YOLO-style architectures remain the dominant family of real-time detectors, especially in resource-constrained environments such as electronic-warfare receivers, where latency and computational footprint are critical. Since our objective is to assess whether multi-resolution processing improves real-time TF detection, YOLO-based models offer the most relevant and practically deployable comparison points.
>
> That said, our evaluation does not rely exclusively on YOLO models. Both the initial and revised versions include a Transformer-based detector (RT-DETR) as an additional baseline. RT-DETR was selected because it is one of the few Transformer architectures designed for low-latency inference. However, as illustrated in the revised Figure 4, RT-DETR remains substantially heavier than the YOLO-based models, making it less suitable for embedded platforms.
>
> **3. Statistical variation and training noise**
>
> Including variance across multiple training runs is difficult in our setting, as the full dataset (private + public versions) is approximately 1 TB and each training run requires considerable computation. For this reason, we were unable to report standard deviations across multiple independent trainings.
>
> However, two factors mitigate this limitation:
>
> - The dataset is extremely large, reducing sampling variability and training noise.
> - The performance gains of MRS-YOLO are highly consistent across model sizes, SNR levels, and evaluation metrics.
>
> This consistency makes it unlikely that the improvements arise from stochastic variation.
>
> **4. Justification of the methodology over other alternatives**
>
> Section 3.3 has been fully rewritten to better articulate why the proposed blocks are well suited to time–frequency representations and to clarify the intuition behind their design. The ablation study shows that both TF-Attn and SCSA consistently improve performance over simpler alternatives, supporting the architectural choices behind MRS-YOLO.
>
> **5. Runtime analysis and fairness of comparisons**
>
> We added an appendix table with FLOPs, parameter counts, and measured inference latencies for all evaluated models.
>
> **6. Removal of “LLM usage disclosure”**
>
> This has been addressed.
>
> **7. Transferability to unseen waveform types and real-world robustness**
>
> The revised version includes an appendix section (*Discussion on the Use of Simulated Datasets*) that directly addresses these questions.

---

### Official Review · Reviewer_5pyn · 2025-11-01

**Soundness:** 2
**Presentation:** 1
**Contribution:** 1
**Rating:** 2
**Confidence:** 4

**Summary:**

MRS-YOLO is the method introduced by the authors for TF based signal detection. The use of TF and segmentation for detection is not new, and it feels like baselines are missing. The use of multiple TF with various hparams is also widely known and done in practice to compensate with the limitation of each individual TF tradeoff. Lastly, the paper proposes an empirical validation but on a synthetic dataset designed for the purpose of the empirical validation of this submission

** Strength **
- being able to do segmentation and annotation of time series is a relevant problem

** Weakness **
- Many typos such as `summerize`, and some sentences could use some shortening rewording to make it easier to read
- why is some part of the text blue?
- Too impractical experimental details to be reproducible. The authors provide some information that feel useless such as the PyThon version but no other required information to reproduce results
- the major weakness is in using synthetic data where there is no clear ablation and justification making the entire empirical results likely bias to this design. This also gives near 0 insight about the transferability of the method to the real world

**Strengths:**

See summary

**Weaknesses:**

See summary

**Questions:**

See summary

---

> ### Author Response · Authors · 2025-11-26
> **To Assigned Reviewer 5pyn**
>
> We thank the reviewer for the numerous remarks, which will help improve the quality and clarity of the manuscript. We appreciate the time dedicated to providing detailed feedback.
>
> We agree with the reviewer that multi-resolution time–frequency analysis is not new. Many works have indeed explored this question, typically for classification or binary detection on a single acquisition. However, to the best of our knowledge, and unless the reviewer has a specific reference in mind, there is no prior work addressing joint temporal–frequency object detection in multi-resolution spectrograms. Our contribution lies in adapting a modern detection architecture to this setting and demonstrating that multi-resolution fusion significantly improves object-level detection performance.
>
> In the revised version, all typos and formatting issues reported by the reviewer have been corrected, and several sections have been rewritten for improved clarity and readability.
>
> The reviewer highlights a legitimate reproducibility concern. This has now been addressed: we provide a fully reproducible experimental pipeline based on a publicly released dataset (excluding sensitive electronic-warfare waveforms). The repository includes a simulator allowing users to regenerate the dataset and training protocol, ensuring transparent and verifiable reproduction of the results.
>
> The reviewer also raises an important question regarding the realism and transferability of simulated data. We emphasize that the underlying simulator is detailed and physically grounded: it models interference, overlapping emissions, Doppler effects, realistic rise/fall times, and waveform libraries whose characteristics match documented specifications. The acquisition chain is based on an actual system. Simulation-based evaluation is standard practice in electronic-warfare research, where real datasets are rarely accessible and almost never labeled, yet detection models are routinely trained on simulated data before deployment.
>
> That said, we fully agree that validation on real-world data would further strengthen the contribution. After a thorough investigation of available RF datasets, we found that existing public corpora are not suitable for multi-resolution time–frequency object detection: they lack precise bounding-box annotations (carrier frequency, duration, bandwidth), do not provide raw 1D signals required for generating multi-resolution STFTs, and are not sufficiently diverse or challenging to justify the use of a multi-resolution deep-learning model. We remain fully open to any dataset suggestions the reviewer might provide.
>
> Finally, in response to this concern, we have added a dedicated section in the appendix (“Discussion on the Use of Simulated Datasets”) that clarifies the role of simulated data in this work, the constraints specific to electronic-warfare applications, the limitations associated with simulation, and why the use of simulated datasets remains justified and standard in this domain.

---

### Official Review · Reviewer_TfcY · 2025-11-04

**Soundness:** 2
**Presentation:** 3
**Contribution:** 3
**Rating:** 2
**Confidence:** 3

**Summary:**

The paper presents MRS-YOLO, a multi-resolution spectrogram-based signal detection method. The method's major contribution is combining two network-based technologies 1) YOLO, a popular framework from computer vision that has demonstrated fast and accurate object detection, 2) time-frequency attention, specifically using the approach of separating convolutions in the time and frequency. These approaches are implemented in a multi-resolution setting where 5 separate STFTs using different window widths are used as the inputs.

**Strengths:**

The paper is well written and easy to follow. The authors clearly indicate their awareness of relevant literature and the benchmarking and ablation studies do a good job of clearly outlining the advantages and tradeoffs of their design choices.

**Weaknesses:**

My biggest complaint about the benchmarking is the authors' use of non-public data as the only benchmark. Without a clear understanding of the data and its properties it is impossible to make a clear assessment of the model's advantages or anticipate its failure modes. Having said that if the authors could repeat their benchmarking with a well-conceived simulated data example it would substantially improve the paper and I would be willing to increase my score.

**Questions:**

There appears to be blue text at the top of section 3. This should be corrected.

---

> ### Author Response · Authors · 2025-11-26
> **To Assigned Reviewer TfcY**
>
> We thank the reviewer for the careful evaluation and for acknowledging the quality of our literature review, benchmarking, and ablation studies. We appreciate the recognition of the methodological effort invested in the design and analysis of MRS-YOLO.
>
> We fully understand that the main concern relates to the use of a private dataset as the sole benchmark. While we maintain that evaluating the method on this dataset is meaningful—because it reflects real electronic-warfare detection scenarios where signals are extremely difficult to detect and classify—we agree that reproducibility is a legitimate requirement. In such operational contexts, low- and mid-SNR conditions are critical, and we believe that demonstrating that MRS-YOLO outperforms conventional baselines in these regimes is itself an application-driven contribution.
>
> Nevertheless, to directly address the reviewer’s concern, the revised version of the paper now includes results on a publicly released RF dataset (Dataset B) that we have generated and made available for reproducibility. This dataset contains a broad variety of telecommunications waveforms and conventional radar signals. While it naturally excludes classified electronic-warfare emitters, it provides a fully documented simulation pipeline, precise detection annotations, and multi-resolution STFT representations. We believe that releasing this simulator and dataset constitutes an additional scientific contribution, especially since existing public RF datasets are typically not designed for low-SNR detection, do not provide bounding-box annotations, and rarely support multi-resolution training.
>
> To ensure timely results during the review process, we initially generated a 40,000-sample version of this dataset. The revised manuscript now incorporates results from a full 100,000-sample version, yielding statistically stable performance curves.
>
> Finally, we thank the reviewer for pointing out the formatting issue (“blue text” at the beginning of Section 3). This has been corrected.

---

### Author Response · Authors · 2025-11-26
**Response to All Reviewer**

Dear Reviewers,

We thank you for your careful reading and constructive comments. They have led to a substantially revised version of the paper, which we believe significantly improves both clarity and scientific value.

Across the reviews, we identified three main axes of criticism:

1. Reproducibility of the dataset and results.
2. Reliance on simulated RF data and lack of evaluation on real-world or non-RF datasets.
3. Presentation issues (typos and some formatting problems).

Below we summarize how the revised manuscript addresses these points and guide you to the main changes.

---

## 1. Reproducibility: public simulator and detailed implementation

To address concerns about reproducibility and the use of non-public data:

- We have created a fully reproducible, publicly available dataset and simulator:

  An anonymized repository is now available at:
  **https://github.com/ICLRanonymous2026/ICLR2026DataSimulator/**

  It contains all components needed to regenerate our new synthetic RF-like dataset (“Dataset B”), including signal models, noise/interference models, and labeling procedures (bounding boxes, classes, and SNR configurations).

- We have re-run the complete experimental suite on the public dataset. Figure 2 and Table 2 have been fully recomputed.

- We have substantially rewritten the *Training setup and implementation details* paragraph (Section 3.4 in the revised version) to make experiments easier to reproduce, in line with Reviewer 5pyn’s concerns.

Within the GitHub organization above, reviewers can exactly reproduce datasets B and C, models, and training.

---

## 2. Synthetic vs. real data, and evaluation beyond RF signals

### (a) Use of simulated data

We thank the reviewers for raising concerns about the exclusive use of simulated RF data. As detailed in the new appendix *Discussion on the Use of Simulated Datasets*, this choice is driven by the electronic warfare (EW) context: real RF datasets with raw 1D signals, dense time–frequency bounding boxes, multi-resolution STFT support, multiple classes, and sufficient scale for deep learning simply do not exist publicly.

We performed a systematic search across RF, audio, and bioacoustic datasets, and none satisfied these requirements simultaneously. In EW practice, training on simulated data and deploying models on non-public real recordings is standard, and we now state this explicitly in the revised manuscript.

### (b) Generality beyond RF

To evaluate whether MRS-YOLO remains useful outside RF applications, we introduced a third dataset composed of synthetic frequency-coded signals (*Dataset C*). These signals are abstract time–frequency patterns built from controlled frequency trajectories and silence structures and are not tied to any RF propagation assumptions.

Dataset C is designed such that:

- Some classes are indistinguishable in a **single-resolution** STFT.
- The same classes become separable when **multiple resolutions** are available.

This makes Dataset C an application-agnostic test of whether multi-resolution fusion provides tangible benefits beyond RF-specific characteristics.

As reported in the revised Section 3.5, MRS-YOLO shows consistent improvements over all single-resolution YOLO baselines on this dataset, both in recall and in class disambiguation. In particular, classes purposely confounded at any single window size become distinguishable in the multi-resolution architecture. This confirms that the benefits of MRS-YOLO arise from the multi-resolution structure itself and are not artifacts of a particular RF simulation pipeline.

---

## 3. Writing, formatting, and equation organization

We thoroughly revised the manuscript to improve clarity and address all reported formatting issues:

- Corrected typographical errors and awkward formulations (including explicitly noted ones such as “summerize”).
- Removed unintended formatting artifacts, including the blue text at the beginning of Section 3.
- Following Reviewer p72W’s recommendation, Section 3.3 has been completely rewritten in a more intuitive and concise manner. Detailed mathematical derivations have been moved to the appendix.
- The discussion on processing times has been relocated to the appendix, where it is presented as a full benchmark.

---

We hope that these substantial revisions address the main concerns regarding reproducibility, dataset design, and presentation.

---

> ### Author Response · Authors · 2025-11-26
> **Guide to the main textual and structural changes**
>
> For convenience, we summarize the main revisions:
>
> ### Abstract
> - Improved explanation of the relationship between the Heisenberg uncertainty principle and spectrogram resolution.
> - Strengthened motivation for multi-resolution detection.
>
> ### Introduction
> - Corrected typos and improved justification for using multiple STFT window lengths.
>
> ### Related Work
> - Light revision of “Multi-resolution Signal Representations” to better position the contribution.
>
> ### Method
> - Fixed all formatting issues in Section 3.
> - Entirely rewrote Section 3.3 in a more compact, conceptual form; derivations moved to the appendix.
> - Expanded Section 3.4 to present datasets A, B, and C and improved implementation details.
>
> ### Experiments (Section 3.5)
> - Added results for new public dataset B and the non-RF frequency-code dataset C.
> - Moved processing-time analysis to the appendix.
>
> ### Section 3.6
> - Minor wording improvements.
>
> ### Appendix
> - Added detailed inference-time measurements.
> - Relocated mathematical derivations from the main text, as suggested.

---

### Author Response · Authors · 2025-12-03
**Clarification on the Timing of the Revised Manuscript and Discussion Interruption**

Following the program chairs’ decision to revert reviews to their pre-discussion state due to the leak incident, we understand that reviewer scores cannot be updated and that no further reviewer discussion was possible. We acknowledge the necessity of this procedure given the circumstances.

In this context, we wish to clarify that our revised manuscript incorporates substantial changes directly addressing all reviewer concerns. These modifications required significant effort within the review timeline: creation of a new publicly reproducible dataset and simulator, rerunning all experiments on this dataset, introduction of an additional non-RF benchmark (Dataset C), restructuring of major methodological sections, and an improved inference-time analysis. The revised paper is therefore markedly different from the original submission.

Because the exhaustive revision of the manuscript was posted on 26/11, and the discussion phase was interrupted on 28/11, reviewers had only a very short window in which to see the updated version and did not have a realistic opportunity to evaluate it or adjust their assessments. This timing effectively prevented the intended iterative exchange.

We note that the extent and depth of the revisions, together with the full reproducibility now provided, could not be reflected in updated reviewer assessments due to the abrupt end of the discussion phase. We hope that this context will be taken into account in the overall evaluation.

---

### Note · Program_Chairs · 2026-01-17
**Submission Desk Rejected by Program Chairs**

The following references in this submission do not refer to real documents and/or have major errors in bibliographic information:

 Yifan Li and Lei Zhou. Slnet: Attention-based multi-resolution spectrogram fusion. In Proc. IEEE INFOCOM, pp. 1112-1120, 2023.